

# Novel attractive pairing interaction
# in strongly correlated superconductors

**Priyo Adhikary and Tanmoy Das***

Department of Physics, Indian Institute of Science, Bangalore, Karnataka 560012

* tnmydas@iisc.ac.in

## Abstract

Conventional and unconventional superconductivity, respectively, arise from attractive (electron-phonon) and repulsive (many-body Coulomb) interactions with fixed-sign and sign-reversal pairing symmetries. Although heavy-fermions, cuprates, and pnictides are widely believed to be unconventional superconductors, recent evidence in one of the heavy fermion superconductor ($CeCu_2Si_2$) indicate the presence of a novel conventional type pairing symmetry beyond the electron-phonon coupling. We present a new mechanism of attractive potential between electrons, mediated by emergent boson fields (vacuum or holon) in the strongly correlated mixed valence compounds. In the strong coupling limit, localized electron sites are protected from double occupancy, which results in an emergent holon fields. The holon states can, however, attract conduction electrons through valence fluctuation channel, and the resulting doubly occupied states with local and conduction electrons condenseas Cooper pairs with onsite, fixed-sign, s-wave pairing symmetry. We develop the corresponding self-consistent theory of superconductivity, and compare the results with experiments. Our theory provides a new mechanism of superconductivity whose applicability extends to the wider class of intermetallic/mixed-valence materials and other flat-band metals.


# 1  Introduction

Superconductivity arises from the formation of electron-electron pairs, namely, Cooper pairs. Celebrated Bardeen-Cooper-Schrieffer (BCS) theory showed that an effective attractive potential between electrons can emanate from the electron-phonon coupling, resulting in a fully gapped, constant sign superconducting (SC) gap (conventional $s$-wave symmetry). [1] Interestingly, discussions of unconventional superconductivity from repulsive interactions dates back to 1965. [2] It was shown that Cooper pairs can be formed in a repulsive interaction medium, provided the corresponding gap function changes sign in the momentum space [2–5]. The first heavy-fermion (HF) superconductor CeCu$_2$Si$_2$ [6] was widely believed to be an unconventional superconductor. [7–10] Subsequently, more HF superconductors, [11] followed by cuprate, and pnictide superconductors are discovered to feature unconventional pairings with either nodal $d$-wave, or nodeless but sign-reversal $s^{\pm}$-pairing symmetry, or their various irreducible combinations. [12]

However, the pairing symmetry, and the pairing mechanism in the first-discovered heavy-fermion compound CeCu$_2$Si$_2$ are recently called into questions. Earlier reports of nuclear quadrupole resonance (NQR) data revealed a $T^3$ behavior in the relaxation rate without a coherence peak, suggesting the presence of line nodes in the SC gap structure. [13–15] Observation of four-fold modulation in the upper critical field $H_{c2}$ in CeCu$_2$Si$_2$ can predict a point-node $d$-wave pairing state [16], provided the Fermi surface (FS) anisotropy is small enough to cause the same modulation. [17] Finally, the observation of a spin resonance in the SC state by inelastic neutron scattering measurement [18] can be interpreted as to arise from sign-reversal of the SC gap if the resonance peak is very sharp and its energy lies within the SC gap amplitude. More recently, counter-evidence of fully gapped superconductivity are obtained in various measurements including point-contact tunneling spectroscopy, [19, 20] specific heat, [21–23] magnetic penetration depth, [23, 24] and thermal conductivity [23]. The field-angle dependence of the specific heat data also shows no evidence of gap anisotropy. [22] Furthermore, the observed robustness of superconductivity to disorder supports the absence of sign-reversal in the pairing symmetry scenario. [23, 25] These results collectively signal towards a conventional, fixed-sign, isotropic pairing symmetry in CeCu$_2$Si$_2$.

CeCu$_2$Si$_2$ has an interesting phase diagram exhibiting two SC domes under pressure, with an antiferromagnetic (AFM) quantum critical point (QCP) lying beneath the first SC dome, while a valence fluctuation critical point is possibly present at the second dome. [26–28] The proximity to the AFM QCP inspires the proposals of spin-fluctuation mediated unconventional, sign-changing pairing symmetry. [24, 29, 30] The valence fluctuation, which is ubiqui-

tous in HF compounds, can promote superconductivity with unconventional pairing mechanism. [8,9,26,27,31,32] In particular, it is widely argued by various groups that the vertex correction due to valence-fluctuation exchange can directly mediate a pairing channel, [9,31,32] or can augment pairing strength arising from other sources [33,34]. Kondo coupling can induce various unconventional pairings. [10,35–40] Following the overwhelming evidence of conventional pairing symmetry, the electron-phonon coupling problem with strong Coulomb interaction is revisited recently. [33,41,42] In general, electron-phonon coupling, if present, can be overturned by the strong onsite Coulomb repulsion in the HF quasiparticles exhibiting effective mass $\sim 10^3$ times the bare mass.

Our present work is motivated by the question: Can there be other source of attractive potential for superconductivity in general? Here, we provide a new mechanism of attractive potential originating from the interplay between the Coulomb interaction and valence fluctuations. The physical picture is illustrated in Fig. 1. When the Coulomb interaction is strong on the $f$-electron's site, double $f$-electron's occupancy is prohibited. Within the field theory view, a singly occupied $f$-electron site is annexed with an unoccupied $f$-state — a bosonic holon field — which repels another $f$-electron to occupy the state. However, the unoccupied $f$-site can be occupied by a conduction electron since the presence of valence fluctuation channel allows mutation between the $f$- and conduction electrons. Remarkably, we show here that the doubly occupied state with $f$- and conduction electrons *condense* like a Cooper pair. Mathematically, as we integrate out the boson fields (unoccupied holons), we obtain a robust, new *attractive* potential channel between the conduction electrons and singly occupied $f$-sites, naturally commencing onsite, constant sign, $s$-wave like superconductivity. Conceptually, this process is somewhat analogous to the theory of meson mediated attractive nuclear force, except here the attraction commences between onsite electrons. We formulate the corresponding theory of superconductivity, and find excellent agreement with the recently observed fully gap, constant sign gap features in CeCu$_2$Si$_2$, [19–25] as well as in the Yb-doped CeCoIn$_5$ superconductors [43]. We predict a definite relationship between SC $T_c$ and valence fluctuation (coherence) temperature $T_K$, and other unique properties of the present theory, which are amenable to verifications.

## 2 Theory

The low-energy phenomena of HF compounds are well described by the periodic Anderson impurity (PAI) model [44,45], which has four parts:

$$H = \sum_{\mathbf{k},\sigma} \xi_{\mathbf{k}} c_{\mathbf{k}\sigma}^{\dagger} c_{\mathbf{k}\sigma} + \xi_f \sum_m f_m^{\dagger} f_m + \sum_{\mathbf{k},\sigma,m} v_{\mathbf{k}} c_{\mathbf{k}\sigma}^{\dagger} f_m + U \sum_m f_m^{\dagger} f_m f_{-m}^{\dagger} f_{-m} + \text{h.c.}, \quad (1)$$

where $c_{\mathbf{k}\sigma}^{\dagger}$ ($c_{\mathbf{k}\sigma}$) is the creation (annihilation) operator for the conduction electron with spin $\sigma = \pm 1/2$. The conduction electron has a dispersion $\xi_{\mathbf{k}}$, with $\mathbf{k}$ being crystal momentum. The strongly correlated $f$-electrons are treated as impurity, sitting on each unit cell with an onsite potential $\xi_f$. The valence fluctuations between the conduction and correlated electrons lead to a hybridization potential $v_{\mathbf{k}}$. Finally, $f$-electrons are subjected to a strong Hubbard interaction $U$. (The model also holds for narrow 'band' $f$-electrons as long as $U \gg D_f$, with $D_f$ being its bandwidth.) Such a model is well studied in the literature, and can be projected onto the Kondo-lattice model using a Schrieffer-Wolf transformation [46]. Another popular route to solve this problem is the so-called slave-boson approach. [47–51]

The basic phenomenologies of the slave-boson model can be described in two parts. A single $f$-orbital on a given site has four Fock states, namely, doubly occupied site ($d$), singly occupied site ($\bar{f}_m$), and unoccupied site ($e$). Clearly, $d$ and $e$ operators are bosons, while $\bar{f}_m$

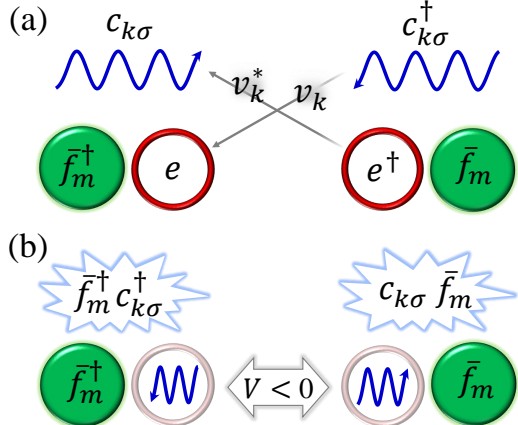

Figure 1: Illustration of the valence fluctuation mediated attractive potential. (a) The unoccupied state (holon) in each valence fluctuation term can attract another conduction electron through the valence fluctuation channel. The conjugate process also occurs simultaneously. Wavy lines depict conduction electrons ($c, c^\dagger$), while filled ($\bar{f}, \bar{f}^\dagger$)) and open ($e, e^\dagger$) circles give singly occupied and unoccupied $f$-sites, respectively. Bar symbol over $f$-operators emphasizes that they are single-$f$-electrons occupied states. Arrows dictate valence fluctuation channels. (b) As we integrate out the unoccupied states ($e, e^\dagger$), we obtain an effective interaction $V < 0$, forming Cooper pair between the single site $\bar{f}$-electron and conduction $c$ electron.

are fermions, with $m$ being the spin index (owing to spin-orbit coupling, $m$ can, in general, have many multiplets). In the $U \to \infty$ limit where double occupancy is strictly prohibited, one can project out the $d$-states.[1] The $f$-orbitals can be expressed in the remaining three Fock states as $f_m = e^\dagger \bar{f}_m$ with the constraint $Q \equiv n_e + n_{\bar{f}} = 1$, where $n_e = e^\dagger e$, $n_{\bar{f}} = \sum_m \bar{f}_m^\dagger \bar{f}_m$ are the corresponding number density at every site. [47–49, 51, 52] Hence we obtain,

$$H = \sum_{\mathbf{k},\sigma} \xi_{\mathbf{k}} c_{\mathbf{k}\sigma}^\dagger c_{\mathbf{k}\sigma} + \bar{\xi}_f \sum_m \bar{f}_m^\dagger \bar{f}_m + \omega_e e^\dagger e + \sum_{\mathbf{k},\sigma,m} \left( v_{\mathbf{k}} c_{\mathbf{k}\sigma}^\dagger e^\dagger \bar{f}_m + v_{\mathbf{k}}^* \bar{f}_m^\dagger e c_{\mathbf{k}\sigma} \right). \qquad (2)$$

We have introduced a onsite potential $\omega_e > 0$ for the unoccupied state, which arises as a Lagrangian multiplier to conserve the number of $f$-electron states to $Q = 1$ in the $U \to \infty$ limit. $\omega_e$ is considered to be site-independent, respecting the translational invariance, which physically implies that all holons are condensed to the same energy. The renormalized $\bar{f}$-electron's energy is $\bar{\xi}_f = \xi_f + \omega_e = Z\xi_f$, where the corresponding band renormalization factor $Z$ is defined as $Z = 1 + \eta$ with $\eta = \omega_e/\xi_f$.

Eq. (2) is our starting point in this work. This is not exactly solvable due to the presence of the $e, e^\dagger$-states. Popular methods involve hard-core boson model (classical), or mean-field theory around the saddle point of $\langle e \rangle$ [49, 53, 54]. Here we include the quantum fluctuations of the holons, and solve Eq. (2) within the quantum field theory approach.

The last term in Eq. (2) implies that each valence fluctuation process generates (or annihilates) a boson field $e^\dagger$ ($e$), whose job is to prohibit double occupancy on the $f$-sites. However, the unoccupied states or holons can attract another conduction electron (and vice versa), i.e., they trigger another valence fluctuation process. The two valence fluctuations process can be tied together to generate an effective interaction potential, which turns out to be attractive at low-energy. Mathematically, this is done by integrating out the coherent bosonic $e$,

---

[1]Our result also holds for finite interaction strength as long as U is larger than the bandwidth of the f-orbitals. In addition, even when double occupancy is not strictly ruled out, and that one includes the d-states, the same pairing interaction is obtained. In this case, however additional terms arise from the d-states which requires numerical methods for solutions.

$e^\dagger$-operators to obtain an effective interaction potential $V_{\mathbf{kk'}}(i\omega_n)$. Sparing the details to Appendix A, we present the final result of an effective interacting Hamiltonian (in the static limit) as

$$H_{\text{eff}} = \sum_{\mathbf{k},\sigma} \xi_{\mathbf{k}} c^\dagger_{\mathbf{k}\sigma} c_{\mathbf{k}\sigma} + \bar{\xi}_f \sum_m \bar{f}^\dagger_m \bar{f}_m + \sum_{\mathbf{kk'},\sigma\sigma',mm'} V_{\mathbf{kk'}} \, c^\dagger_{\mathbf{k}\sigma} \bar{f}_m \bar{f}^\dagger_{m'} c_{\mathbf{k'}\sigma'}. \tag{3}$$

Spin conservation leads to $\sigma + m = \sigma' + m'$. The most impressive aspect of the above result lies in the form of the effective potential

$$V_{\mathbf{kk'}}(i\omega_n) = v_{\mathbf{k}} v^\dagger_{\mathbf{k'}} \frac{2\omega_e}{(i\omega_n)^2 - \omega_e^2}, \tag{4}$$

where $i\omega_n$ is the bosonic Matsubara frequency. In what follows, in the low energy limit $i\omega_n < \omega_e$ and $\omega_e > 0$ (since holon's energy is generally positive), Eq. (4) produces an *attractive* potential. This is one of our principle results of this work. As in the case of the BCS theory, [1] we consider here the static limit $i\omega_n \to 0$ limit, yielding

$$V_{\mathbf{kk'}} = -\frac{2v_{\mathbf{k}} v^\dagger_{\mathbf{k'}}}{\omega_e} < 0. \tag{5}$$

For a generic attractive potential, the pair correlation function has a logarithm divergence with temperature (see Appendix C), and we have a SC ground state. Looking at Eq. (3), we find that the Cooper pairs form here between the conduction electron and singly occupied $\bar{f}_m$-site with the SC gap parameter defined as

$$\Delta_{\mathbf{k}} = \frac{2v_{\mathbf{k}}}{\omega_e} \sum_{\mathbf{k'}} v^\dagger_{\mathbf{k'}} \langle c_{\mathbf{k'}\sigma} \bar{f}_m \rangle. \tag{6}$$

Here we make few observations. (i) This is an inter-band pairing between the spin-$\frac{1}{2}$ conduction electron and single-site $f$-electron with $m$ multiplet. (ii) The $\mathbf{k}$—dependence of the SC gap is solely determined by that of the hybridization term $v_{\mathbf{k}}$ in Eq. (5). (iii) This is a finite-momentum pairing, but unlike the Fulde-Ferrel-Larkin-Ovchinnikov state (FFLO) or the pair density wave state, here the Cooper pair solely absorbs the conduction electron's momentum. (For dispersive, narrow $f$-band, which is often the case in many HF systems, Cooper pairs can have zero center-of-mass momentum.) (iv) The SC state, in general, does not have the particle-hole symmetry, unless at $\xi_{\mathbf{k}} = \bar{\xi}_f$. (v) Symmetry of the Cooper pairs is dictated by the values of $m$, $\sigma$, and the parity of $V_{\mathbf{kk'}}$. In CeCu$_2$Si$_2$, the hybridization occurs between the Ce-$f$ and Ce-$d$ orbitals of the same Ce-atom, [30] and thus the hybridization potential can be considered as onsite, i.e., $v_{\mathbf{k}} = v$. For onsite hybridization, one expects a spin-singlet pair for $m = \pm 1/2$ (or higher order antisymmetric spin component if $|m| > 1/2$). For an attractive potential, spin-singlet, onsite ($s$-wave) pairing state has the highest eigenvalue as obtained in the BCS case as well. [1]

## 3 Mean-field results and critical phenomena

So far, we have obtained all results exactly. We now invoke the mean-field theory for superconductivity. The effective mean-field Hamiltonian reads

$$H_{\text{MF}} = \sum_{\mathbf{k}\sigma} \xi_{\mathbf{k}} c^\dagger_{\mathbf{k}\sigma} c_{\mathbf{k}\sigma} + \bar{\xi}_f \sum_m \bar{f}^\dagger_m \bar{f}_m + \sum_{\mathbf{k}\sigma m} \Delta_{\mathbf{k}} \bar{f}^\dagger_m c^\dagger_{\mathbf{k}\sigma} + \text{h.c.}. \tag{7}$$

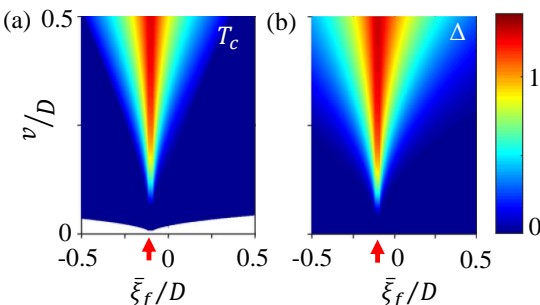

Figure 2: SC phase diagram with respect to valence fluctuation potential $v$ and renormalized $f$-electron's energy $\bar{\xi}_f$. (a), The SC transition temperature $T_c$ is plotted in the $(v, \bar{\xi}_f)$ space, scaled with respect to the conduction electron's bandwidth $D$. We set $\xi_f/D = -0.1$. The white region for small values of $v$ gives the SC-forbidden region (Eq. (11)). (b), SC gap amplitude $\Delta$ (at $T = 0$) plotted in the same parameter space. Above the critical value of $v$, both $T_c$ and $\Delta$ grows with $v^2$ as in Eq. (9). Interestingly, optimal superconductivity commences at a finite value of $\bar{\xi}_f$ where all the holon fields condense to $\omega_e \to 0$, and the pairing potential $V \to \infty$.

The corresponding self-consistent gap equation is (see Appendix B)

$$\Delta_{\mathbf{k}} = \frac{2v_{\mathbf{k}}}{\omega_e} \sum_{\mathbf{k'}} v_{\mathbf{k'}}^* \frac{\Delta_{\mathbf{k'}}}{4E_{0\mathbf{k'}}} \sum_{\nu=\pm} \nu \, \tanh\left(\frac{\beta E_{\mathbf{k'}}^{\nu}}{2}\right). \tag{8}$$

$\nu = \pm$ are the two quasiparticle bands: $E_{\mathbf{k}}^{\pm} = \xi_{\mathbf{k}}^- \pm E_{0\mathbf{k}}$, where $E_{0\mathbf{k}} = \sqrt{(\xi_{\mathbf{k}}^+)^2 + |\Delta_{\mathbf{k}}|^2}$, and $\xi_{\mathbf{k}}^{\pm} = (\xi_{\mathbf{k}} \pm \bar{\xi}_f)/2$. $\beta = 1/k_B T$.

In the case of onsite hybridization $v_{\mathbf{k}} = v$, the $\mathbf{k}$-dependence of the pairing potential is removed. This gives $V_{\mathbf{kk'}} = -\frac{2|v|^2}{\omega_e}$ with $\omega_e > 0$, leading to a 'conventional' $s$-wave pairing symmetry $\Delta_{\mathbf{k}} = \Delta$. Taking advantage of the onsite attractive potential, and $s$-wave pairing channel, we can solve Eq. (8) analytically. Solutions of Eq. (8) in the two asymptotic limits of $T \to 0$, and $\Delta \to 0$ yield the gap amplitude $\Delta$ and $T_c$ as

$$\begin{aligned}
\Delta &= \bar{D} e^{-\frac{1}{2\lambda}} \left[1 + r e^{-\frac{1}{\lambda}}\right]^{1/2}, \\
k_B T_c &= D_\gamma e^{-\frac{1}{\lambda}} \left[1 - \left(\frac{\bar{\xi}_f}{2D_\gamma}\right)^2 e^{\frac{2}{\lambda}}\right]^{1/2}.
\end{aligned} \tag{9}$$

Here $\bar{D} = \sqrt{D^2 - \bar{\xi}_f^2}$, $D_\gamma = 2D\gamma/\pi$ and $r = (D + \bar{\xi}_f)/(D - \bar{\xi}_f)$, with $\gamma$ being the Euler constant, and $D = 1/2N$, and $N$ are bandwidth and density of states of conduction electrons at the Fermi level. The SC coupling constant is defined as

$$\lambda = \frac{2N|v|^2}{\omega_e} = 2|\eta|^{-1} N J_{\mathrm{K}}, \tag{10}$$

where $J_{\mathrm{K}} = |v|^2/|\xi_f|$ is the Kondo coupling constant. $\eta$ is defined below Eq. (2). The first terms before the parenthesis in both $\Delta$ and $T_c$ are the usual BCS solutions, while the correction terms within the parenthesis have important consequences. The correction term in Eq. (9) suggests that superconductivity arises above a critical value of the coupling constant

$$\frac{1}{\lambda} < \ln\left(\frac{2D_\gamma}{|\bar{\xi}_f|}\right). \tag{11}$$

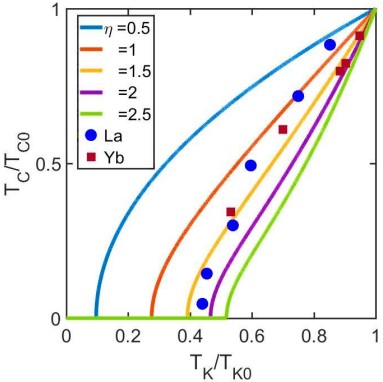

Figure 3: Relationship between $T_c$ and $T_K$. We demonstrate the relationship between $T_c$ and $T_K$ for several values of the exponent $\eta$ (from Eq. (13)). Interestingly, $T_c$ vanishes below some critical value of $T_K$, where the cutoff value decreases with decreasing $\eta$. $T_c$, $T_K$ are normalized to some highest values of $T_{c0}$, $T_{K0}$, respectively, for each values of $\eta$. For CeCoIn$_5$, Yb and La dopings [55] are known to modulate the valence fluctuation strength $T_K$, giving an intriguingly similar $T_c$ versus $T_K$ relationship, as predicted by our theory in Eq. (13). Experimental values agree well for $\eta \sim 1-1.5$ for $\bar{\xi}_f = 0.7$eV.

This implies that there exists a lower critical value of the hybridization $\nu_c$ above which superconductivity is possible. Since $\nu$ is related to the coherence temperature $T_K$, we show below that the above constraint translates into a lower limit for $T_K$ to produce superconductivity. This result is in contrast to the BCS result where any infinitesimal electron-phonon coupling is sufficient for finite $T_c$. Interestingly, the BCS ratio $\Delta/k_B T_c$ is not a universal constant here, even in the weak coupling limit. In the limit of $D \gg \bar{\xi}_f$, we recover BCS-type behavior of $\Delta \to De^{-1/2\lambda}$, and $k_B T_c \to D_\gamma e^{-1/\lambda}$, with $\Delta/k_B T_c \to 1.73 e^{1/2\lambda}$, suggesting a strong coupling limit of the superconductivity.

Plots of $\Delta$ and $T_c$ as a function of $\nu$ and $\bar{\xi}_f$ are shown in Fig. 2. Both phase diagrams exhibit funnel like behavior in the $\nu - \bar{\xi}_f$ space. We highlight here two key features. (i) In $T_c$ plot we find a white region for small values $\nu$ which marks the forbidden (non-SC) region dictated by the constraint $1/\nu^2 > (N/2\omega_e) \ln |2D_\gamma/\bar{\xi}_f|$ (Eq. (11)). In the rest of the regions where both $\Delta$ and $T_c$ are finite, we obtain a second order phase transition with the critical exponent of $1/2$. (ii) Secondly, superconductivity is optimal at a characteristic value of $\bar{\xi}_f \neq 0$ (marked by arrows in Fig. 2). At this point $\omega_e \to 0$ ($\bar{\xi}_f = \xi_f$) and hence the pairing potential $V \to \infty$, stipulating maximum superconductivity. At the optimal $T_c$, $f$-electron's band renormalization $Z \to 1$.

## 3.1 Connection to coherence temperature $T_K$

From Eq. (4), it is evident that $\omega_e$ is analogous to the Debye frequency of the electron-phonon mechanism. The essential dependence of $T_c$ and $\lambda$ on observable parameters such as coherence temperature $T_K$ can be derived using the saddle point approximation [49,53,54]. For this case, Eq. (2) can be solved exactly, [56] yielding $k_B T_K = De^{-1/NJ_K}$. Therefore, from Eq. (10), we find that the SC coupling constant $\lambda$ depends on $T_K$ as

$$\frac{1}{\lambda} = \eta \ln\left(\frac{D}{k_B T_K}\right). \tag{12}$$

This result is consistent with the fact that the Kondo critical point prompts optimal superconductivity as obtained in CeCu$_2$Si$_2$, [26] as well as in many other HF superconductors.

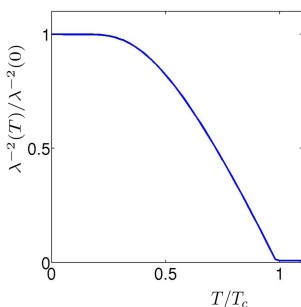

Figure 4: Computed superfluid density as a function of temperature. The temperature dependence shows a typical exponential behavior at low-$T$ as seen in CeCu$_2$Si$_2$.

[8, 9, 11, 57–59] However, $T_c$ is terminated below a critical $T_K$ which can be obtained from Eq. (9) as

$$(k_B T_c)^2 = D_\gamma^2 \left( \frac{k_B T_K}{D} \right)^{2|\eta|} - \frac{\bar{\xi}_f^2}{4},$$ (13)

where $\eta$ is the same as before. Eq. (13) is another important result of our theory, which finds a surprisingly consistent agreement with experimental data (see Fig. 3). We plot $T_c$ and $T_K$ for several parameter values in Fig. 3. Both the critical behavior and the power-law dependence between $T_c$ and $T_K$ agree remarkably well with the experimental data of La, and Yb doped CeCoIn$_5$ samples. [55]

## 4 Signatures of pairing structure

### 4.1 Meissner effect

Unlike the typical Cooper pair of two conduction electrons with opposite momenta in other types of superconductors, here we have a pairing between conduction electron and correlated singly occupied $f$-electrons. The conduction electrons directly couple to the gauge field $\mathbf{A}$ as $\mathbf{p}' = \hbar\mathbf{k} - \frac{e}{c}\mathbf{A}$. On the other hand, the $f$-states do not couple to the vector potential in its localized limit. Importantly, despite that the magnetic field couples only to the conduction electron, we find a complete exclusion of the magnetic field at $T \to 0$, a hallmark of superfluid state. Interestingly, however, in the strongly localized limit of the $f$-orbitals, the Meissner effect experiments will exhibit charge of the Cooper pair to be $-e$, instead of $-2e$ as in other Conventional Cooper pair between two itinerant electrons. Caution to be taken in realistic heavy-fermion systems, where the band structure calculation [29] shows weak dispersion of the $f$-electrons, which couple to the external gauge field, and hence may contribute to the Cooper pair charge of $-2e$ or a value between $-e$ to $-2e$ on average.

Here we proceed with computation of the diamagnetic ($\mathbf{J}_d$) and paramagnetic ($\mathbf{J}_p$) current of the conduction electrons only:

$$\mathbf{J}_d = \frac{e^2 \mathbf{a}}{c} \sum_{\mathbf{k}\sigma} \frac{1}{m_\mathbf{k}} c_{\mathbf{k}\sigma}^\dagger c_{\mathbf{k}\sigma}, \quad \mathbf{J}_p = e \sum_{\mathbf{k}\sigma} \mathbf{v}_\mathbf{k} c_{\mathbf{k}\sigma}^\dagger c_{\mathbf{k}\sigma}.$$ (14)

$\mathbf{v}_\mathbf{k}$ and $m_\mathbf{k}$ are the velocity and effective mass, respectively, of the conduction electron, and $\mathbf{a}$ is the Fourier component of the vector potential $\mathbf{A}$. Using the mean-field solution of the quasiparticle bands, the superfluid density (inversely proportional to the magnetic penetration

depth) is obtained to be

$$\lambda_{ij}^{-2}(T) = \frac{4\pi e^2}{c^2} \sum_{\mathbf{k}}' \left[ \frac{1}{m_{ij,\mathbf{k}}} \left( 1 - \sum_{\nu} (\alpha_{\mathbf{k}}^{\nu})^2 \tanh\left(\frac{\beta E_{\mathbf{k}}^{\nu}}{2}\right) \right) \right.$$
$$\left. - \frac{\beta}{2} v_{i\mathbf{k}} v_{j\mathbf{k}} \sum_{\nu} (\alpha_{\mathbf{k}}^{\nu})^2 \mathrm{sech}^2\left(\frac{\beta E_{\mathbf{k}}^{\nu}}{2}\right) \right], \tag{15}$$

$\nu = \pm$ for two quasiparticle bands. [Prime symbol over the summation indicates that it is restricted within the first quadrant of the Brillouin zone, since both $+\mathbf{k}$ and $-\mathbf{k}$ fermions are included exclusively to obtain Eq. (15) (see Appendix D).] $(\alpha_{\mathbf{k}}^{\mp})^2 = \frac{1}{2}\left(1 \mp \frac{\xi_{\mathbf{k}}^+}{E_{0\mathbf{k}}}\right)$ is the coherence factors of the mean-field solutions. The numerical evaluation of Eq. (15) yields an exponential behavior of superfluid density as $T \to 0$, as shown in Fig. 4. This behavior is also observed experimentally in CeCu$_2$Si$_2$ [23, 24] as well as in Yb-doped CeCoin$_5$ [43].

## 4.2 Spin-resonance mode

For unconventional pairing symmetry, the sign-reversal of the SC gap leads to a spin-resonance mode at $\omega_{\mathrm{res}} \leq 2\Delta$. [12] Such a mode is rather weak in intensity and may lie above $2\Delta$ for conventional (fixed sign) pairing symmetry. [60] Experimentally, a resonance is observed in the SC state in CeCu$_2$Si$_2$ at $\mathbf{Q} \sim (0.215, 0.215, 1.458)$ in r.l.u. in the energy scale of $\sim 0.2$ meV which is roughly at $4k_B T_c$ ($T_c \sim 0.6$ K). [18]

The present pairing symmetry has few interesting collective spin modes which can explain the above experimental behavior. For the calculation of spin fluctuation to be tractable we consider that the $f$-electrons possess spin $m = \pm 1/2$. In this case, the total spin operator can be defined as a summation over conduction spin and $f$-electrons spin:

$$\mathbf{S}_{\mathbf{q}} = \frac{1}{2}\left( \sum_{\mathbf{k}\alpha\beta} c_{\mathbf{k}\alpha}^{\dagger} \boldsymbol{\sigma}_{\alpha\beta} c_{\mathbf{k}+\mathbf{q}\beta} + \sum_{\alpha\beta} \bar{f}_{\alpha}^{\dagger} \boldsymbol{\sigma}_{\alpha\beta} \bar{f}_{\beta} \right). \tag{16}$$

$\alpha$, $\beta$ are spin indices. The transverse spin susceptibility is defined as $\chi(\mathbf{q}, \tau) = \langle T_{\tau} S^+(\mathbf{q}, \tau) S^-(-\mathbf{q}, 0)\rangle$. Solving in the mean-field SC state, we obtain

$$\chi(\mathbf{q}, i\omega_n) = \sum_{\mathbf{k}} \sum_{\mu,\nu=\pm} A_{\mathbf{k}\mathbf{q}}^{\mu\nu} \frac{f(E_{\mathbf{k}+\mathbf{q}}^{\mu}) - f(E_{\mathbf{k}}^{\nu})}{i\omega_n + E_{\mathbf{k}}^{\nu} - E_{\mathbf{k}+\mathbf{q}}^{\mu}}, \tag{17}$$

where

$$A_{\mathbf{k}\mathbf{q}}^{\mu\nu} = \frac{1}{2}\left( 1 \pm \frac{\xi_{\mathbf{k}}^+ \xi_{\mathbf{k}+\mathbf{q}}^+ + \Delta_{\mathbf{k}} \Delta_{\mathbf{k}+\mathbf{q}}}{E_{0\mathbf{k}+\mathbf{q}} E_{0\mathbf{k}}} \right), \tag{18}$$

$\mu$, $\nu = \pm$ are the band indices, and $\pm$ in Eq. (18) corresponds to amplitude of the oscillators for $\mu = \nu$ (intra-) and $\mu \neq \nu$ (inter-) quasiparticle band transition. Eq. (17) can give various collective excitations, depending on the band structure details. We are here interested in the possible modes inside the SC gap. Indeed, we find the solution of a localized spin-excitation in the SC state at a wavevector which corresponds to the condition $\xi_{\mathbf{k}}^+ = -\xi_{\mathbf{k}+\mathbf{Q}}^+$. (Note that this is not the condition of the conduction electron's FS nesting). In this case, we have a resonance at an energy

$$\omega_{\mathrm{res}} = E_{\mathbf{k}+\mathbf{Q}}^+ - E_{\mathbf{k}}^- \sim \frac{2\Delta^2}{|\bar{\xi}_f|}, \tag{19}$$

in the limit of $\Delta \gg \xi_{\mathbf{k}}^+$. The corresponding oscillator strength of the resonance mode is $A_{\mathbf{k}\mathbf{q}}^{\mu,\nu\neq\mu} = (\xi_{\mathbf{k}}^+)^2 / E_{0\mathbf{k}}^2 > 0$. Since $\bar{\xi}_f > \Delta$, the resonance occurs inside the SC gap, as observed experimentally in CeCu$_2$Si$_2$ [18].

## 4.3 Other measurements

The present theory of valence fluctuation mediated attractive pairing channel can be verified in multiple ways. For example, the present theory predicts a unique Andreev reflection behavior. In a typical normal metal and superconductor interface, as an electron tunnels from the metal into the superconductor side, it reflects back a hole, and vice versa. In our present case, the conduction electron from the normal metal forms a Cooper pair with a $f$-state in the SC sample, and thus *reflects a f-electron to the normal metal*, which can be easily probed. The reflection probably is inversely proportional to the effective mass of the $f$-electron. This means in the limit of the localized $f$-electron case, the Andreev reflection can be strongly suppressed or absent. A suppression of Andreev reflection amplitude is observed in $CeCoIn_5$, [61] and $CeCu_2Si_2$ [19, 20].

As also mentioned in the above section, in the limit of fully localized $f$-orbitals when the coupling to the external gauge field is suppressed, one may find evidence of $-e$ charge of the Cooper pair in such experiments. However, the band structure effect of the $f$-orbitals can help coupling of the $f$-orbitals to the gauge field and hence the charge of the Cooper pair on average can be observed to be somewhere between $-e$ to $-2e$ in experiments.

## 5 Discussions and conclusions

Our theory demonstrates the existence of an attractive pairing potential mediated by the interplay between Coulomb interaction and valence fluctuations. The origin of the attractive potential is the emergent boson field (holon) associated with single-site $f$-states to restrict double occupancy due to strong Coulomb interaction. The effective interaction is a result of multiple valence fluctuations: The holon field generated in a given valence fluctuation is annihilated in the second valence fluctuation, and the resulting two valence fluctuation processes generate an effective interaction between the $f$- and conduction electrons. The interaction is attractive at low-frequency and isotropic in the case of onsite valence fluctuation process. The onsite, attractive interaction naturally gives an isotropic, constant sign $s$-wave pairing channel between the single-site $f$-electrons, and conduction electrons.

Our result of fixed-sign, isotropic $s$-wave pairing channel is consistent with numerous experimental data discussed in the introduction. [19–25] The exponential temperature dependence of point-contact tunneling spectroscopy, [19,20] specific heat, [21–23] thermal conductivity [23], and penetration depth [23, 24] are naturally explained within our model. Moreover, there have been several recent evidence of two-band superconductivity in $CeCu_2Si_2$. [21,22,24] It was shown that most of the above data, as well as the $T^3$ dependence of the NQR data [13–15] can be fitted well with a two-band model with a simple $s$-wave pairing symmetry. This is fully consistent with our theory which has a two-band (conduction and local) behavior with $s$-wave pairing. Furthermore, the proposed pairing (Eq. (6)) is a finite momentum pairing in the limit of fully localized $f$-electrons, and itinerant conduction electrons. Consistently, there have been recent evidence of finite momentum pairing state in $CeCu_2Si_2$. [62] Finally, strong suppression of Andreev reflection amplitude in $CeCoIn_5$, [61] and $CeCu_2Si_2$ [19, 20] are well known, suggesting the involvement of the localized $f$-orbitals in the Cooper pairs.

In addition, the present theory can also explain the other three experimental signatures which were taken earlier as evidence of unconventional, sign-reversal pairing symmetry. (i) The $T^3$ dependence of the NQR relaxation rate $1/T_1$ below $T_c$ in $CeCu_2Si_2$ is often considered as evidence of line nodes in the SC gap structure. [13–15] As mentioned above, a two-band model with purely $s$-wave gap, as in the present case, is shown to reproduce the same power-law behavior of $1/T_1$ without invoking gap nodes. [21,22] Therefore, we anticipate our theory is equally applicable here. (ii) The four-fold angular modulation of $H_{c2}$ in $CeCu_2Si_2$ [16]

can be a signature of the SC gap anisotropy. However, it was shown in a realistic two-band model that a strong anisotropy in $H_{c2}$ (as well as in other quantities) can well arise solely from the Fermi surface anisotropy even for a purely isotropic $s$-wave SC gap. [17] Indeed, the conduction electron's Fermi surface is known to be substantially anisotropic in $CeCu_2Si_2$. [29, 30] (iii) Finally, it is known that a spin-resonance as measured by inelastic neutron scattering experiments can arise either from unconventional, sign-reversal pairing symmetry, or even for a fixed-sign $s$-wave pairing. [60] For sign-reversal pairing gap, the spin-resonance is typically very sharp and its energy must follow $\omega_{\mathrm{res}} < 2\Delta$, where $\Delta$ is the SC gap amplitude. On the other hand, for fixed-sign, conventional pairing, the resonance is usually very broad, and its energy lies at $\omega_{\mathrm{res}} \geq 2\Delta$. The measured spin-resonance in $CeCu_2Si_2$ [18] is indeed quite broad, and the present data cannot discern if the resonance energy lies below or above $2\Delta$. Moreover, our theory also predicts a novel resonance mode at an energy (Eq. (19)) determined by $2\Delta^2/\bar{\xi}_f$.

We compare and contrast the concepts of the present theory with the prior theories of 'conventional' pairing solutions in $CeCu_2Si_2$. Valence fluctuation mediated or assisted pairing mechanism has been a steady theme of discussions in the heavy-fermions community. [8, 9, 26, 27, 31–34] Miyake and Onishi [31, 32] have proposed a phenomenological pairing vertex formula with the help of an empirical valence fluctuation susceptibility defined near its critical point. Unlike our case, the pairing vertex in Ref. [31] does not invoke electron-electron correlation, however, the pairing interaction is argued to be retarded when correlation is included. On the other hand, in our case, the pairing interaction is microscopically derived from the interplay between correlation and valence fluctuation and has a robust solution of attractive channel at the low-energy limit. Our pairing interaction can be considered as a generalized, dynamical Kondo interaction. If we express the interaction in Eq. (3) in terms of local spin and conduction spin interaction, then $V_{\mathbf{kk}'}(\omega)$ can be cast as dynamical Kondo interaction $J_K(\omega)$ (similar result in the static limit can be obtained within the Schrieffer-Wolf transformation [46]). Starting from Kondo interaction with $J_K < 0$, a composite Cooper pair theory was proposed where conduction electron pairs up the (chargeless) fermionic representation of the local spin. [36, 38] Such composite pairing channel is also $s$-wave like in the limit of local Kondo channel. A prior quantum Monte Carlo simulation of periodic Anderson model showed the existence of a $s$-wave pairing interaction. [35] This gives a validation of the attractive pairing interaction we derive in Eq. (4). Finally, we propose that a future dynamical mean-field theory (DMFT) calculation will be valuable to further confirm the existence of the attractive paring solution in such a model.

Finally, we make few remarks about the future extension of the present theory. A full, self-consistent treatment of $T_c$, $\eta$, and $T_K$ requires an Eliashberg-type formalism. Since $T_c$ is significantly low in HF compounds, the present mean-field treatment is however a good approximation for the estimates of $T_c$. The theory also holds for dispersive $f$-electrons state as long as the corresponding bandwidth is much lower than $U$. For a dispersive $f$-state, one can obtain a zero center-of-mass momenta Cooper pair $\langle c_{\mathbf{k}\sigma}^{\dagger} \bar{f}_{-\mathbf{k}m}^{\dagger} \rangle$. Therefore, the present theory is applicable to the wider class of intermetallic and mixed valence superconductors where a narrow band and a conduction band coexist, and possesses finite interband tunneling (valence fluctuation) strength. [63] Our calculation does not include Coulomb interaction between the conduction and $f$-electrons (the Falicov-Kimball type interaction). However, it is obvious that such a Coulomb interaction term will lead to a pair breaking correction (e.g $\mu^*$-term), in analogy with the Coulomb interaction correction to the electron-phonon coupling case (the so-called McMillan's formula) [64]. Finally, the vertex correction to the pairing potential can be envisaged, in analogy with the Migdal's theory, to scale as $m/M$, where $m$, and $M$ are the mass of the conduction and $f$-electrons. Since $M \sim 10^3$ in these HF systems, we argue that the vertex correction can be negligible.

## Acknowledgements

We thank M. B, Maple, T. V. Ramakrishnan, G. Baskaran, B. Kumar, F.D.M. Haldane for discussions and numerous suggestions. The work is supported by the Science and Engineering Research Board (SERB) of the Department of Science & Technology (DST), Govt. of India for the Start Up Research Grant (Young Scientist), and also benefited from the financial support from the Infosys Science foundation under Young investigator Award.

## A  Field theory treatment of the hole states and effective attractive potential

The action of the Hamiltonian in Eq. (2) is broken into four components

$$S = S_c + S_{\bar{f}} + S_e + S_v, \tag{20}$$

where

$$\mathcal{S}_c = \int d\tau \sum_{\mathbf{k},\sigma} \tilde{c}_{\mathbf{k}\sigma}(\tau)(\partial_\tau + \xi_{\mathbf{k}})c_{\mathbf{k}\sigma}(\tau), \tag{21}$$

$$\mathcal{S}_{\bar{f}} = \int d\tau \sum_m \tilde{\bar{f}}_m(\tau)(\partial_\tau + \bar{\xi}_f)\bar{f}_m(\tau), \tag{22}$$

$$\mathcal{S}_e = \int d\tau \, \tilde{e}(\tau)(\partial_\tau + \omega_e)e(\tau), \tag{23}$$

$$\mathcal{S}_v = \int d\tau \sum_{\mathbf{k},\sigma,m} \left( v_{\mathbf{k}}\tilde{c}_{\mathbf{k}\sigma}(\tau)\tilde{e}(\tau)\bar{f}_m(\tau) + \text{h.c.} \right). \tag{24}$$

Here $\tilde{e}, e$ are bosonic coherent states and $\tilde{\bar{f}}, \bar{f}, \tilde{c}, c$ are Grassmann variables for singly occupied $f$-states, and conduction electrons respectively ('tilde' means conjugation). $\tau$ is imaginary time axis. Thermodynamic properties of the system can be calculated from the partition function $\mathcal{Z} = \text{Tr} e^{-\mathcal{S}}$, where the trace is taken over all degrees of freedom of the system. We obtain an effective action $\mathcal{S}_{\text{eff}}$ by integrating out the bosonic variables $\tilde{e}, e$ as

$$
\begin{aligned}
\mathcal{Z} &= \int \mathcal{D}[\tilde{c},c]\mathcal{D}[\tilde{\bar{f}},\bar{f}]\mathcal{D}[\tilde{e},e]e^{-\mathcal{S}_c-\mathcal{S}_{\bar{f}}-\mathcal{S}_e-\mathcal{S}_v}, \\
&= \int \mathcal{D}[\tilde{c},c]\mathcal{D}[\tilde{\bar{f}},\bar{f}]e^{-\mathcal{S}_c-\mathcal{S}_{\bar{f}}} \int \mathcal{D}[\tilde{e},e]e^{-\mathcal{S}_e-\mathcal{S}_v}, \\
&= \int \mathcal{D}[\tilde{c},c]\mathcal{D}[\tilde{\bar{f}},\bar{f}]e^{-\mathcal{S}_{\text{eff}}[\tilde{c},c,\tilde{\bar{f}},\bar{f}]}, 
\end{aligned}
\tag{25}
$$

where

$$S_{\text{eff}} = \mathcal{S}_c + \mathcal{S}_{\bar{f}} - \ln \int \mathcal{D}[\tilde{e},e]e^{-\mathcal{S}_e-\mathcal{S}_v}. \tag{26}$$

It is easier to perform the $\tau$ integration in the Matsubara frequency space. The Fourier transformation to the Matsubara frequency domain of the $e(\tau)$ variable gives $e(\tau) = \frac{1}{\sqrt{\beta}}\sum_n e_n \exp(-i\omega_n\tau)$, where $i\omega_n$ is bosonic Matsubara frequency and $e_n = e(i\omega_n)$. In the Matsubara space, we get

$$\mathcal{S}_e = -\sum_n \tilde{e}_n(\mathcal{G}^e)^{-1}(i\omega_n)e_n, \tag{27}$$

where $\mathcal{G}^e$ is the bare Green's function for the $e_n$-states: $(\mathcal{G}^e)^{-1} = i\omega_n - \omega_e$.

Next we define a bosonic hybridization field $\rho_{\mathbf{k}\sigma m}$ as

$$\rho_{\mathbf{k}\sigma m}(\tau) = \tilde{c}_{\mathbf{k}\sigma}(\tau)\bar{f}_m(\tau), \tag{28}$$

whose Fourier component is $\rho_{\mathbf{k}\sigma m}(\tau) = \frac{1}{\sqrt{\beta}}\sum_n \rho_{\mathbf{k}\sigma m,n}\exp(-i\omega_n\tau)$, where $\rho_{\mathbf{k}\sigma m,n} = \rho_{\mathbf{k}\sigma m}(i\omega_n)$ with $i\omega_n$ being the bosonic Matsubara frequency. Hence we can express the hybridization action as

$$\begin{aligned}
\mathcal{S}_v &= \int_0^\beta d\tau \sum_{\mathbf{k},\sigma,m}\left(v_{\mathbf{k}}\tilde{e}(\tau)\rho_{\mathbf{k}\sigma m}(\tau) + v_{\mathbf{k}}^*\tilde{\rho}_{\mathbf{k}\sigma m}(\tau)e(\tau)\right) \\
&= \sum_{\mathbf{k},\sigma,m}\sum_n\left(v_{\mathbf{k}}\tilde{e}_n\rho_{\mathbf{k}\sigma m,n} + v_{\mathbf{k}}^*\tilde{\rho}_{\mathbf{k}\sigma m,n}e_n\right).
\end{aligned} \tag{29}$$

Interestingly, now in Eqs. (27),(29) the integration over $\tau$-variable is replaced with summation over discrete Matsubara frequencies $n$. Let us say at a given temperature we have $N$ number of Matsubara frequencies. So we define a bosonic spinor $\mathbf{E} = (e_1, e_2, ..., e_N)^T$, and $\tilde{\mathbf{E}} = (\tilde{e}_1, \tilde{e}_2, ..., \tilde{e}_N)$. Similarly, we define a vector for the hybridization field as $\mathbf{V} = (\mathfrak{v}_1, \mathfrak{v}_2, ..., \mathfrak{v}_N)^T$, $\tilde{\mathbf{V}} = (\tilde{\mathfrak{v}}_1, \tilde{\mathfrak{v}}_2, ..., \tilde{\mathfrak{v}}_N)$ where $\mathfrak{v}_n = \sum_{\mathbf{k}\sigma m}v_{\mathbf{k}}\rho_{\mathbf{k}\sigma m,n}$, and $\tilde{\mathfrak{v}}_n = \sum_{\mathbf{k}\sigma m}v_{\mathbf{k}}^*\tilde{\rho}_{\mathbf{k}\sigma m,n}$. Finally, we define a diagonal matrix $\mathbf{G}^{-1}$ for the inverse Green's function $(\mathcal{G}^e)^{-1}$ in Eq. (27), whose components are $\mathbf{G}_{nn}^{-1} = (\mathcal{G}_e)^{-1} = i\omega_n - \omega_e$. Hence we can express Eqs. (27),(29) respectively as

$$\begin{aligned}
\mathcal{S}_e &= -\ \tilde{\mathbf{E}}\cdot\mathbf{G}^{-1}\cdot\mathbf{E}, \tag{30} \\
\mathcal{S}_v &= \ \tilde{\mathbf{E}}\cdot\mathbf{V} + \tilde{\mathbf{V}}\cdot\mathbf{E}. \tag{31}
\end{aligned}$$

Therefore, the last term of Eq. (26) can be evaluated as

$$\int \mathcal{D}[\tilde{\mathbf{E}},\mathbf{E}]e^{-\mathcal{S}_e - \mathcal{S}_v} = \pi^N \det\mathbf{G}^{-1}e^{-\left[\tilde{\mathbf{V}}\cdot\mathbf{G}^{-1}\cdot\mathbf{V}\right]} \tag{32}$$

(We ignored some irrelevant constant factors). The factor of the exponent on the right hand side of Eq. (32) can now be evaluated rigiously. In $T \to 0$ limit, the Matsubara frequencies span from $n = -\infty$ to $\infty$. Hence we obtain,

$$\begin{aligned}
&\tilde{\mathbf{V}}\cdot\mathbf{G}^{-1}\cdot\mathbf{V} \\
&= -\sum_{\substack{\mathbf{k},\sigma,m \\ \mathbf{k}',\sigma',m'}}\sum_{n=-\infty}^{\infty}v_{\mathbf{k}}^*\tilde{\rho}_{\mathbf{k}\sigma m,n}\frac{1}{-i\omega_n + \omega_e}v_{\mathbf{k}'}\rho_{\mathbf{k}'\sigma'm',n} \\
&= \sum_{\substack{\mathbf{k},\sigma,m \\ \mathbf{k}',\sigma',m'}}\sum_{n=0}^{\infty}v_{\mathbf{k}}^*v_{\mathbf{k}'}\frac{2\omega_e}{(i\omega_n)^2 - \omega_e^2}\tilde{\rho}_{\mathbf{k}\sigma m,n}\rho_{\mathbf{k}'\sigma'm',n} \\
&= \sum_{\substack{\mathbf{k},\sigma,m \\ \mathbf{k}',\sigma',m'}}\sum_{n=0}^{\infty}V_{\mathbf{k}\mathbf{k}'}\tilde{\bar{f}}_m(i\omega_n)c_{\mathbf{k},\sigma}(i\omega_n)\tilde{c}_{\mathbf{k}',\sigma'}(i\omega_n)\bar{f}_{m'}(i\omega_n).
\end{aligned} \tag{33}$$

In the last equation, we have substituted the hybridization field into fermionic field from Eq. (28). The effective potential is

$$V_{\mathbf{k}\mathbf{k}'} = v_{\mathbf{k}}^*v_{\mathbf{k}'}\frac{2\omega_e}{(i\omega_n)^2 - \omega_e^2}. \tag{34}$$

## B Mean-field solutions

We use the Nambo-Gorkov basis $\psi_{\mathbf{k}} = (c_{\mathbf{k}\sigma} \quad \bar{f}_m^\dagger)^T$, in which the mean-field Hamiltonian (Eq. (7)) reads

$$H_{\text{MF}}(\mathbf{k}) = \xi_{\mathbf{k}}^- I_{2\times 2} + \xi_{\mathbf{k}}^+ \sigma_z - \Delta_{\mathbf{k}} \sigma_x, \tag{35}$$

where $\sigma_i$ are the $2 \times 2$ Pauli matrices and $I_{2\times 2}$ is a unit matrix. $\xi_{\mathbf{k}}^\pm = (\xi_{\mathbf{k}} \pm \bar{\xi}_f)/2$. The BdG eigenvalues are

$$E_{\mathbf{k}}^\pm = \xi_{\mathbf{k}}^- \pm E_{0\mathbf{k}}, \quad \text{with} \quad E_{0\mathbf{k}} = \sqrt{(\xi_{\mathbf{k}}^+)^2 + |\Delta_{\mathbf{k}}|^2}. \tag{36}$$

The Bogoliubov operators for the two eigenvalues $E_{\mathbf{k}}^\pm$ are

$$\begin{pmatrix} \phi_{\mathbf{k}}^+ \\ (\phi_{\mathbf{k}}^-)^\dagger \end{pmatrix} = \begin{pmatrix} \alpha_{\mathbf{k}}^+ & -\alpha_{\mathbf{k}}^- \\ \alpha_{\mathbf{k}}^- & \alpha_{\mathbf{k}}^+ \end{pmatrix} \begin{pmatrix} c_{\mathbf{k}\sigma} \\ \bar{f}_m^\dagger \end{pmatrix}, \tag{37}$$

where

$$(\alpha_{\mathbf{k}}^\mp)^2 = \frac{1}{2}\left(1 \mp \frac{\xi_{\mathbf{k}}^+}{E_{0\mathbf{k}}}\right). \tag{38}$$

Evaluating the self-consistent gap equation from Eq. (6), we get Eq. (8).

### B.1 Transition temperature $T_c$

For the attractive potential, onsite pairing is more favorable. Hence we set $V_{\mathbf{k}\mathbf{k}'} = -2|v|^2/\omega_e$. In this case, superconducting transition temperature $T_c$ can be obtained by taking the limits of $\Delta \to 0$, which renders $E_{\mathbf{k}}^+ \to \xi_{\mathbf{k}}$, $E_{\mathbf{k}}^- \to -\bar{\xi}_f$, $E_{0\mathbf{k}} \to \frac{|\xi_{\mathbf{k}} + \bar{\xi}_f|}{2}$. From Eq. (8) we obtain

$$1 = \lambda \int_{-D}^{D} \frac{d\xi}{2(\xi + \bar{\xi}_f)} \left[\tanh\left(\frac{\beta_c \xi}{2}\right) + \tanh\left(\frac{\beta_c \bar{\xi}_f}{2}\right)\right], \tag{39}$$

where we have substituted $\lambda = 2N|v|^2/\omega_e$. $\beta_c = 1/k_B T_c$. The first integral in Eq. (39) is a tricky one. In the limit of $D \gg \bar{\xi}_f$, we can approximately evaluate this integral. The first integral of Eq. (39) gives

$$I_1 \approx \lambda \ln\left[\frac{2D_\gamma}{\sqrt{\bar{\xi}_f^2 + (2k_B T_c)^2}}\right], \tag{40}$$

where $D_\gamma = 2D\gamma/\pi$ with $\gamma = 1.78$ being the Euler constant. The second integral is trivial to evaluate which gives

$$I_2 = \lambda \tanh\left(\frac{\beta_c \bar{\xi}_f}{2}\right) \ln\left|\frac{D + \bar{\xi}_f}{-D + \bar{\xi}_f}\right|. \tag{41}$$

In the limit of $D > \bar{\xi}_f$, $I_2 \to 0$. Therefore, we are left with $I_1 = 1$, which gives,

$$(k_B T_c)^2 = D_\gamma^2 e^{-2/\lambda} - \frac{\bar{\xi}_f^2}{4}, \tag{42}$$

Eq. (8) in the main text is obtained from the above equation.

## B.2    SC gap amplitude

Next we take the $T \to 0$ limit in Eq. (8). In this limit, we get $\tanh(\frac{\beta E_{\mathbf{k}}^{\pm}}{2}) \to \pm 1$. Hence we are left with

$$
\begin{aligned}
1 &= \lambda \int_{-D}^{D} \frac{d\xi}{\sqrt{(\xi + \bar{\xi}_f)^2 + 4\Delta^2}} \\
&= \lambda \ln\left( \frac{\sqrt{(D + \bar{\xi}_f)^2 + 4\Delta^2} + D + \bar{\xi}_f}{\sqrt{(D - \bar{\xi}_f)^2 + 4\Delta^2} - D + \bar{\xi}_f} \right) \\
&\approx \lambda \ln\left( \frac{2(D + \bar{\xi}_f)}{\sqrt{(D - \bar{\xi}_f)^2 + 4\Delta^2} - D + \bar{\xi}_f} \right).
\end{aligned}
\tag{43}
$$

In the last equation above, we assumed $D \gg \Delta$. Solving Eq.(43)

$$
\Delta = \bar{D} e^{-\frac{1}{2\lambda}} \left[ 1 + r e^{-\frac{1}{\lambda}} \right]^{1/2},
\tag{44}
$$

where $\bar{D} = \sqrt{D^2 - \bar{\xi}_f^2}$, and $r = (D + \bar{\xi}_f)/(D - \bar{\xi}_f)$. In the weak coupling limit $\lambda \to 0$, we get $\Delta \to \bar{D} e^{-\frac{1}{2\lambda}}$ (notice the factor of $2\lambda$ in the exponent) while in the strong coupling limit, we obtain the BCS-type formalism of $\Delta \to \sqrt{D^2 + \bar{\xi}_f^2} e^{-\frac{1}{\lambda}} \approx D e^{-\frac{1}{\lambda}}$.

## C    Pair susceptibility

To affirm that there exists a pairing instability in Eq. (3) in the main text, we compute the pair-pair correlation function. We consider the pair field

$$
b_{\mathbf{k}}(\tau) = \sum_{\sigma, m} c_{\mathbf{k}\sigma}(\tau) \bar{f}_m(\tau),
\tag{45}
$$

where $\tau$ is the imaginary time. The pair susceptibility is defined as

$$
\chi_p(\mathbf{q}, i\omega_n) = \int_0^{\beta} \sum_{\mathbf{k}} \left\langle \mathcal{T}_{\tau} b_{\mathbf{k}}(\tau) b_{\mathbf{k}+\mathbf{q}}^{\dagger}(\tau') \right\rangle e^{-i\omega_n(\tau - \tau')},
\tag{46}
$$

where $\mathcal{T}_{\tau}$ is the time ordered operator. Using Wick's decomposition, we evaluate the above average as

$$
\left\langle \mathcal{T}_{\tau} b_{\mathbf{k}}(\tau) b_{\mathbf{k}+\mathbf{q}}^{\dagger}(\tau') \right\rangle = \sum_{\sigma, m} \mathcal{G}_m^f(\tau - \tau') \mathcal{G}_{\mathbf{k},\sigma}^c(\tau - \tau') \delta_{\mathbf{q},0},
\tag{47}
$$

where $\mathcal{G}_{\mathbf{k},\sigma}^c(\tau - \tau') = \langle \mathcal{T}_{\tau} c_{\mathbf{k}\sigma}(\tau) c_{\mathbf{k}\sigma}^{\dagger}(\tau') \rangle$ is the conduction electron's Green's function, and $\mathcal{G}_m^f(\tau - \tau') = \langle \mathcal{T}_{\tau} \bar{f}_m(\tau) \bar{f}_m^{\dagger}(\tau') \rangle$ is the Green's function for the single site $\bar{f}_m$ states. In the fermionic Matsubara frequency $ip_n$ space these two Green's functions become $\mathcal{G}_{\mathbf{k},\sigma}^c(ip_n) = (ip_n - \xi_{\mathbf{k}})^{-1}$, and $\mathcal{G}_m^f(ip_n) = (ip_n - \bar{\xi}_f)^{-1}$. Substituting the Green's functions in Eq. (46), and doing the Fourier transformation we get

$$
\chi_p(i\omega_n) = \frac{1}{\beta} \sum_{\mathbf{k},\sigma,m} \sum_{n'} \mathcal{G}_m^f(ip_{n'}) \mathcal{G}_{\mathbf{k},\sigma}^c(i\omega_n - ip_{n'}).
\tag{48}
$$

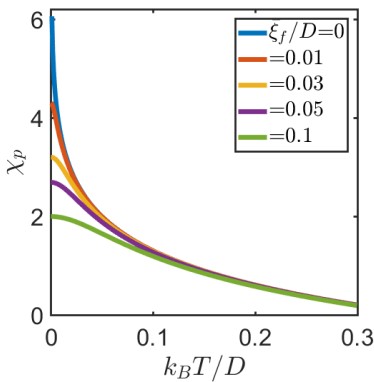

Figure 5: Static pair susceptibility at $\mathbf{q} = 0$ as a function of temperature for different values of $\bar{\bar{\xi}}_f$. As expected from Eq. (51) the pair correlation function diverges at $T \to 0$ for $\bar{\bar{\xi}}_f \to 0$.

Substituting the corresponding Green's functions and performing the standard Matsubara frequency summation on $ip_{n'}$, we arrive at

$$\chi_p(i\omega_n) = \sum_{\mathbf{k}} \frac{1 - f(\bar{\xi}_f) - f(\xi_{\mathbf{k}})}{\bar{\xi}_f + \xi_{\mathbf{k}} - i\omega_n}, \tag{49}$$

$f(\xi)$ is the Fermi distribution function. We are interested in the $\omega \to 0$, and $\mathbf{q} \to 0$ limits. Taking analytic continuation to the real frequency plane $i\omega_n \to \omega + i\delta$, the pair susceptibility becomes

$$\chi_p(\omega \approx 0) = \frac{N}{2} \int_{-D}^{D} d\xi \frac{\tanh(\frac{\beta \bar{\xi}_f}{2}) + \tanh(\frac{\beta \xi}{2})}{\bar{\xi}_f + \xi}. \tag{50}$$

This equation is nothing but the R.H.S. of Eq. (39), except the constant factor $V$. Again in the limit of $D \gg \bar{\xi}_f$ this integral gives the solution as in Eq. (40). Hence we get

$$\chi_p(T) = N \ln \left[ \frac{2D_\gamma}{\sqrt{\bar{\xi}_f^2 + (2k_B T)^2}} \right]. \tag{51}$$

Interestingly, unlike the typical BCS case, the pair correlation function does not have a logarithmic divergence as $T \to 0$ except in the limit of $\bar{\xi}_f \to 0$. This is the reason superconductivity is limited by a minimum limit of the coupling constant $\lambda$ and $T_K$ to overcome the onsite energy $\bar{\xi}_f$ as discussed in the main text.

## D  Further details of the Meissner effect

Unlike the typical Cooper pair of two conduction electrons with opposite momenta in other mechanism, here we have a pairing between conduction electron and correlated singly occupied $f$-electrons. How do these Cooper pairs couple to the applied magnetic field? It is easy to envisage that conduction electrons directly couple to the gauge field $\mathbf{A}$ as $\mathbf{p}' = \hbar \mathbf{k} - \frac{e}{c} \mathbf{A}$. On the other hand, the $f$-states do not couple to the vector potential in its localized limit. Therefore, important changes are expected here in the Meissner effects, compared to typical BCS case.

First of all, under the magnetic field the BdG states become chiral and thus the Bogolyubov states $\phi^{\pm}_{\pm\mathbf{k}}$ and the corresponding eigenvalues $E^{\pm}_{\pm\mathbf{k}}$ for $\pm\mathbf{k}$ are no longer the same. Hence we treat them explicitly as:

$$
\begin{aligned}
c_{\mathbf{k}\sigma} &= \alpha_{\mathbf{k}}\phi^+_{\mathbf{k}} + \beta_{\mathbf{k}}(\phi^-_{\mathbf{k}})^{\dagger} \\
c_{-\mathbf{k}\sigma} &= \alpha_{\mathbf{k}}\phi^+_{-\mathbf{k}} + \beta_{\mathbf{k}}(\phi^-_{-\mathbf{k}})^{\dagger}.
\end{aligned}
\tag{52}
$$

$\alpha_{\mathbf{k}}$, and $\beta_{\mathbf{k}}$ are the coherence factors at zero magnetic field. The corresponding change in the eigenvalue are $E^{\nu}_{\pm\mathbf{k}} = E^{\nu}_{\mathbf{k}} \mp \frac{e}{c}\mathbf{a}.\mathbf{v}_{\mathbf{k}}$, where $\nu = \pm$, and $\mathbf{a}$ is the Fourier component of the vector potential in the momentum space. $\mathbf{v}_{\mathbf{k}} = \partial\xi_{\mathbf{k}}/(\hbar\partial\mathbf{k})$ is the conduction band velocity with $\mathbf{v}_{-\mathbf{k}} = -\mathbf{v}_{\mathbf{k}}$. $E^{\nu}_{\mathbf{k}}$ are the eigenvalues without the magnetic field, and hence $E^{\nu}_{-\mathbf{k}} = E^{\nu}_{\mathbf{k}}$. In the weak magnetic field limit, this corresponds to the change in the Fermi Dirac distribution functions as $f(E^{\nu}_{\pm\mathbf{k}}) = f(E^{\nu}_{\mathbf{k}}) \mp (\frac{e}{c}\mathbf{a}.\mathbf{v}_{\mathbf{k}})\frac{\partial f}{\partial E^{\nu}_{\mathbf{k}}}$. The two current operators are

$$
\mathbf{J_d}(\mathbf{q}) = \frac{e^2}{c}\mathbf{a}(\mathbf{q})\sum_{\mathbf{k}\sigma}{}' \frac{1}{m_{\mathbf{k}}}\left[ c^{\dagger}_{\mathbf{k}-\mathbf{q}\sigma}c_{\mathbf{k}\sigma} + c^{\dagger}_{-\mathbf{k}+\mathbf{q}\sigma}c_{-\mathbf{k}\sigma} \right],
\tag{53}
$$

$$
\mathbf{J_p}(\mathbf{q}) = e\sum_{\mathbf{k}\sigma}{}' \mathbf{v}_{\mathbf{k}-\mathbf{q}}\left[ c^{\dagger}_{\mathbf{k}-\mathbf{q}\sigma}c_{\mathbf{k}\sigma} - c^{\dagger}_{-\mathbf{k}+\mathbf{q}\sigma}c_{-\mathbf{k}\sigma} \right].
\tag{54}
$$

Here $m_{\mathbf{k}}$ is the effective mass of the conduction electron. In the above two equations we utilized the fact that $\mathbf{v}_{-\mathbf{k}} = -\mathbf{v}_{\mathbf{k}}$, and $m_{-\mathbf{k}} = m_{\mathbf{k}}$. The prime over the summation indicate that the summation is restricted to the first quadrant of the Brillouin zone. By substituting Eq. (52) and after a lengthy and straightforward calculation, we arrive at

$$
\mathbf{J_d}(0) = -\frac{e^2\mathbf{a}(0)}{c}\sum_{\mathbf{k}}{}' \frac{1}{m_{\mathbf{k}}}\left[ 1 - (\alpha^+_{\mathbf{k}})^2\,\tanh\left(\frac{\beta E^+_{\mathbf{k}}}{2}\right) - (\alpha^-_{\mathbf{k}})\,\tanh\left(\frac{\beta E^-_{\mathbf{k}}}{2}\right) \right],
\tag{55}
$$

$$
\mathbf{J_p}(0) = \frac{e^2\beta}{2c}\sum_{\mathbf{k}}{}' (\mathbf{a}.\mathbf{v}_{\mathbf{k}})\mathbf{v}_{\mathbf{k}}\left[ (\alpha^+_{\mathbf{k}})\,\mathrm{sech}^2\left(\frac{\beta E^+_{\mathbf{k}}}{2}\right) + (\alpha^-_{\mathbf{k}})\,\mathrm{sech}^2\left(\frac{\beta E^-_{\mathbf{k}}}{2}\right) \right].
\tag{56}
$$

Next we take the linear response theory and within the London's equations, we define the penetration depth $\lambda(T)$ as $\lambda^{-2}_{ij} = -\frac{4\pi}{c}\frac{J_i(0)}{a_j(0)}$, where $\mathbf{J} = \mathbf{J_p} + \mathbf{J_d}$ is the total current. $i, j$ are the spatial coordinates. This gives the final result given in Eq. (15). This equation reduces to the typical BCS form in the case of $\xi_{\mathbf{k}} = -\bar{\xi}_f$.

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
