# Peer review of "Novel attractive pairing interaction in strongly correlated superconductors"

_SciPost Physics, doi:SciPost Phys. 7, 078 (2019)_

## Round 1 · Referee Report · Anonymous (Referee 1) · 2019-7-8

Strengths

1 Clearly written.
2 Well organised structure.
3 Generally interesting result.

Weaknesses

1 Reference to important former theoretical work is missing.
2 Weak connection to experiments.
3 Application to real materials questionable.

Report

This manuscript presents a possible attractive pairing interaction which is discussed to appear in the valence fluctuation regime of heavy fermion materials. The authors use the slave-boson approach to map the Anderson impurity model in the strong correlation limit onto an effective valence fluctuation model where the hybridization is coupled to holons. From this model they derive an unconventional type of pairing in a similar way as it is done with the electron-phonon interaction in the BCS theory. However, the difference is that this mechanism describes a pairing between f- and c-electrons and is therefore directly bound to the valence fluctuations. The authors draw a number of interesting conclusions from that, as for example relation between Tc and Kondo temperature, Meissner effect, and spin resonances.

Indeed, this manuscript presents an interesting new type of superconducting pairing which might have some relevance in the heavy fermion superconductors. The paper is clearly written and the theoretical method and results are discussed in an appropriate way. However, the authors should make the following important points more clear before the manuscript can be recommended for publication:

1.) A similar theory of a valence fluctuation mediated pairing was introduced some time ago by Miyake and coworkers (see for example Physica B 259, 676 (1999)). Among other issues this theory could successfully explain the unusual transport properties in the valence fluctuation regime as for example the drop of the T^2 behavior of the resistivity. The authors completely ignore such important work and they should discuss how their work is related to the pairing interaction introduced there. What is the advance of their approach over former theoretical work.

2.) Former transport measurements (for example Holmes and Jaccard, Physica B: Condensed Matter 378, 339 (2006)) suggest that in the valence fluctuation regime the correlations are rather weak. This is however in contradiction to the considered model where the Coulomb repulsion of the f electrons is set to infinity which leads to prohibition of double occupancy. The authors should comment on this aspect.

3.) Moreover, in the strong correlation regime the Anderson impurity model can be mapped onto the Kondo model where it is known that the f electron energy is always below the Fermi level leading in the case of a lattice to a large Fermi surface where according to the Luttinger theorem the f-spins are counted as electrons. Thus, valence fluctuations are not possible in this regime. How can then the true valence fluctuation physics be captured by the slave boson model which the authors consider? How the authors can be sure that the discussed pairing scenario is assigned to the valence fluctuation superconductivity in heavy fermion materials under pressure?
  • validity: low
  • significance: ok
  • originality: good
  • clarity: high
  • formatting: reasonable
  • grammar: good

Author:  Tanmoy Das  on 2019-09-09  [id 597]

(in reply to Report 1 on 2019-07-08)
Category:
remark
question
answer to question
correction
pointer to related literature

Please find authors response to Referee 1 in the attached pdf file (a separate response file is also provided for referee 2).

Attachment:

Referee_report_referee1_revised_manuscript_for_resubmission.pdf

---

## Round 1 · Referee Report · Anonymous (Referee 2) · 2019-7-13

Strengths

1- An interesting proposal is presented for superconductivity in heavy fermions 2- This is potentially testable in experiments

Weaknesses

1- Poor writing 2- Assumptions need not be discussed/justified a little more strongly 3- Possible experimental tests must be fleshed out

Report

The manuscript discusses a holon-based mechanism for superconductivity in heavy fermion systems. Starting from a Kondo lattice model, the authors introduce a holon-singlon-doublon representation to describe the Hilbert space of each impurity. A strong on-site repulsion on the impurity site prohibits doublons. Subsequently, a chemical potential is introduced for the holons in order to preserve the impurity occupancy. The authors argue that these holons can mediate superconductivity by generating an effective attraction between pairs. Notably, each pair is composed of one conduction and one impurity electron.

The presentation and the writing needs improvement. Spelling errors include “guage” and “Ovbnikov”. There are incorrect usages that make the manuscript difficult to read (e.g., “former materials” after listing three families of materials, using “condensate” as a verb, “primitive CeCu2Si2 compound”, “…incipiently relies on…”, “..intercepting the first dome”, etc.). The authors should revise the manuscript in this regard.

The central idea presented here is interesting and plausibly applicable to CeCu2Si2. However, there are several points that are not clear to me:

a. The authors review experiments on CeCu2Si2. In particular, they point out an apparent contradiction between early indications of nodal unconventional pairing and recent claims of gapped conventional pairing. In their subsequent analysis, they invoke many simplifying assumptions to argue for s-wave gapped pairing. Their stand is not clear here. Do they suggest that CeCu2Si2 is a gapped superconductor? How do they explain earlier observations?

b. The authors refer to the e-particle as a gauge field. In my view, this language is not appropriate. It does not represent a gauge degree of freedom (like photons), even though a gauge structure can possibly be invoked for the representation. It may be better to call it a pseudo-particle in analogy with Schwinger bosons or Abrikosov fermions.

c. There are several hidden and perhaps unnecessary assumptions in Eq. 1.
i. Why do all impurity orbitals (all m’s) have the same energy?
ii. Why is the hybridization, v_k, independent of conduction electron spin? This seems particularly unreasonable. This assumption goes on to give s-wave character in pairing. This needs strong justification.
iii. Later on, the authors assume on-site hybridization to motivate s-wave pairing. They should justify this or atleast provide references to previous studies where this has been argued.

d. The e, f and d particles are assumed to be fermions. This must be clearly specified.

e. I do not find the discussion regarding the relation between T_K and v_k to be convincing, especially in the light of the h-d-e-f representation used here. What is the mean value n_e that is appropriate? When is this justified?

f. Eq. 14 seems to have a typo with regard to the position of \mathbf{a}. It should perhaps sit inside the k summation. What is the significance of the primed summation here?

g. The behaviour shown in Fig. 4 is described as 'exponential decay’ in the text. This is not the commonly understood meaning of `exponential decay’.

h. Fig. 3 compares experimental data for two materials with calculated T_C and T_K. These quantities are calculated using several assumptions that lead to s-wave pairing. Are these assumptions justified in these materials?

i. In my opinion, the discussion section should be expanded to discuss clear experimental signatures of the proposed mechanism.
i. The discussion of Andreev reflection is not fleshed out. There seem to be strong assumptions about the character of the normal side, e.g., its electrons hybridize with the conduction electrons of the heavy fermion, but not with the local moments.
ii. Are there clear ways to distinguish charge e vs. 2e?
iii. The authors state “…we find a complete exclusion of the magnetic field at T->0”. How is this known? How strongly does this depend on the assumptions made (e.g., s-wave pairing)?
  • validity: ok
  • significance: good
  • originality: good
  • clarity: low
  • formatting: perfect
  • grammar: below threshold

Author:  Tanmoy Das  on 2019-09-09  [id 598]

(in reply to Report 2 on 2019-07-13)
Category:
remark
question
answer to question
correction

Please find authors response to Referee 2 in the attached pdf file (a separate response file is also provided for referee 1).

Attachment:

Referee_report_referee2_revised_manuscript_for_resubmission.pdf

---

## Round 2 · Referee Report · Anonymous (Referee 1) · 2019-10-18

Report

In the revised manuscript, the authors made some improvement. In my view, the author's response to the referee reports and the changes in the manuscript are satisfactory. Given the clarification and changes made to the new version, I recommend the publication of this work in SciPost.

---

## Round 2 · Author Response

We have uploaded our responses in pdf file to both referees' reports. Our response files also included our revised manuscript with blue color text highlighting the revisions made here. The same version without the blue color is also uploaded on arxiv (arXiv:1906.01312v2). Since our responses include equations and figures, we are not able to type it on this box.

---

## Round 2 · List of Changes

1) We have expanded the Discussion section (last section below bibliography) 2) We have added new citations in Ref. 31, 32, 52. 3) Few typographical errors are rectified. 4) In several places, few comments are added and/or revised in accordance to referees' suggestions.

(All changes are highlighted in blue color in the revised pdf file appended to each referee reports.)

---

## Editorial Decision

published